# Extracellular Vesicle- and Mitochondria-Based Targeting of Non-Small Cell Lung Cancer Response to Radiation: Challenges and Perspectives

**DOI:** 10.3390/cancers16122235

**Published:** 2024-06-15

**Authors:** Sergey Leonov, Anna Dorfman, Elizaveta Pershikova, Olumide Inyang, Lina Alhaddad, Yuzhe Wang, Margarita Pustovalova, Yulia Merkher

**Affiliations:** 1Department of Cell Technologies, Institute of Future Biophysics, 141700 Dolgoprudny, Russia; 2Department of Cellular Mechanisms of Memory Pathology, Institute of Cell Biophysics, Russian Academy of Sciences, 142290 Pushchino, Russia; 3Biomedical Engineering, Technion-Israel Institute of Technology, Haifa 3200003, Israel

**Keywords:** extracellular vesicles, metastasis, endocytosis, mitochondria transfer, FLASH therapy, pulsed high-power microwave radiation, radioresistance, radiosensitivity

## Abstract

**Simple Summary:**

Radiation therapy stands out as a primary approach for managing individuals with non-small cell lung cancer (NSCLC). Nevertheless, the predominant impediment to achieving successful therapeutic outcomes lies in the resistance exhibited by tumor cells to radiation exposure. Mitochondrial structure abnormality and defects were found to be in high correlation with malignancy and radioresistance. The cytotoxic impact of radiation on cancer cells is most probably dependent on mitochondria; therefore, the exchange of mitochondrial organelles, DNA, or proteins could potentially serve as an effective strategy for modulating their sensitivity to radiation therapy. In this review, we aimed to uncover novel mechanisms for studying NSCLC’s response to radiation.

**Abstract:**

During the cell life cycle, extracellular vesicles (EVs) transport different cargos, including organelles, proteins, RNAs, DNAs, metabolites, etc., that influence cell proliferation and apoptosis in recipient cells. EVs from metastatic cancer cells remodel the extracellular matrix and cells of the tumor microenvironment (TME), promoting tumor invasion and metastatic niche preparation. Although the process is not fully understood, evidence suggests that EVs facilitate genetic material transfer between cells. In the context of NSCLC, EVs can mediate intercellular mitochondrial (Mt) transfer, delivering mitochondria organelle (MtO), mitochondrial DNA (mtDNA), and/or mtRNA/proteinaceous cargo signatures (MtS) through different mechanisms. On the other hand, certain populations of cancer cells can hijack the MtO from TME cells mainly by using tunneling nanotubes (TNTs). This transfer aids in restoring mitochondrial function, benefiting benign cells with impaired metabolism and enabling restoration of their metabolic activity. However, the impact of transferring mitochondria versus transplanting intact mitochondrial organelles in cancer remains uncertain and the subject of debate. Some studies suggest that EV-mediated mitochondria delivery to cancer cells can impact how cancer responds to radiation. It might make the cancer more resistant or more sensitive to radiation. In our review, we aimed to point out the current controversy surrounding experimental data and to highlight new paradigm-shifting modalities in radiation therapy that could potentially overcome cancer resistance mechanisms in NSCLC.

## 1. Introduction

Non-small cell lung cancer (NSCLC) remains a significant global health challenge, and radiation therapy is a cornerstone in its treatment [1]. However, the effectiveness of radiation therapy can be compromised by the emergence of resistance mechanisms within cancer cells [2]. A new area of research that is getting attention is the study of extracellular vesicles (EVs) and mitochondria in influencing how NSCLC responds to radiation therapy. However, due to the complex and heterogeneous nature of NSCLC, coupled with the relatively recent recognition of the pivotal roles played by EVs and mitochondria in radioresistance/radiosensitivity mechanisms within this specific cancer subtype, there is a very limited number of published research works. According to the National Library of Medicine (NLM) (accessed on 10 May 2024), there are only five documented studies that directly explore the interplay between mitochondria or EVs and radioresistance/radiosensitivity in NSCLC, in contrast to the considerably larger body of literature available for other cancer types, totaling 74 published works. 

EVs, including exosomes and microvesicles, are membrane-enveloped vesicles released by various cells, including cancer cells, into the extracellular environment. They have important roles in cell communication by transferring organelles and bioactive molecules like proteins, nucleic acids, and lipids between cells. Recent studies have revealed that EVs play a crucial role in mediating cellular responses to radiation [3,4,5,6]. These findings highlight the significant impact of EVs in modulating radioresistance in cancer cells.

Mitochondria are known as the cell’s powerhouse. They play a role in producing energy, as well as regulating cell death pathways, DNA damage response, and redox balance. Dysfunctional mitochondria have been linked to heightened radioresistance in cancer cells. However, targeting mitochondrial functions has demonstrated significant promise in making cancer cells more susceptible to radiation therapy [7]. Mitochondria have multiple roles in cancer, affecting cancer cell growth, ability to avoid cell death, and spread to other parts of the body. Understanding the molecular mechanisms underlying mitochondrial dysfunction in cancer cells is crucial for developing targeted therapeutic strategies. New techniques have improved how we study mitochondria and their role in cancer [8,9]. New evidence shows that cells can exchange mitochondria in different ways, such as tunneling nanotubes, EVs, and cell fusion. Mitochondrial transfer between cancer cells and surrounding stromal cells has been implicated in promoting tumor growth, metastasis, and therapy resistance [10]. The intricate interplay between cancer cells and their microenvironment underscores the significance of targeting mitochondrial transfer for cancer treatment. This phenomenon highlights the potential of disrupting this process to develop new therapeutic approaches against cancer. Such targeted strategies could lead to improved outcomes for cancer patients, making it a promising area of research in oncology.

By compiling and analyzing the latest advancements in this field, our objective is to explore the role of EVs and mitochondria in various cancer cell responses to radiation. Through this modest review, our primary goal is to uncover potential novel mechanisms that can be rigorously tested regarding NSCLC’s response to radiation. By doing so, we will bridge the existing knowledge gap and stay at the forefront of cutting-edge research in this rapidly advancing field. We will explore the molecular mechanisms underlying the interplay between EVs, mitochondria, and radiation response in NSCLC cells. We will summarize techniques used to extract and characterize mitochondria. We will also discuss how mitochondria can be transferred between different cell types, with potential applications in NSCLC. We will also talk about how targeting EVs and mitochondria can improve radiation therapy for NSCLC patients.

## 2. Modalities in Radiotherapy of Non-Small Cell Lung Cancer (NSCLC)

Today, cancer is the second leading cause of global mortality [11]. Various detection methods, such as blood tests utilizing cancer-specific markers, imaging techniques (including MRI, CT, X-ray, and ultrasound), and endoscopy, are employed to identify cancer. However, metastasis accounts for over 90% of all cancer-related fatalities [12,13]. Predicting and effectively addressing metastasis can significantly impact the overall survival rate [14]. This is especially important in NSCLC, which accounts for about 85% of all lung cancers and is relatively insensitive to chemo- and radiotherapy. 

Roughly 75% of non-small cell lung cancer cases are diagnosed at advanced stages, specifically stages III or IV, according to the American Joint Committee on Cancer (AJCC) staging system [15]. This greatly reduces the chances of curative treatments. Although the five-year survival rate for stage I lung cancer is an encouraging 61.0%, the early detection rate typically lags behind at less than 26% [16]. A significant number of NSCLC patients cannot undergo radical surgery, not only due to the advanced stage of the cancer at diagnosis but also because of their comorbidities, significantly increasing the risk of complications. Predicting and effectively addressing metastasis can significantly impact the overall survival rate [14]. This is especially important in NSCLC, which accounts for about 85% of all lung cancers and is relatively insensitive to chemo- and radiotherapy. Presently, the selection of treatment strategies for NSCLC is contingent upon both the disease stage and the overall health status of the patients. Customizing the treatment is crucial to guaranteeing the utmost effective care, considering the distinct histopathological and molecular traits and cancer stage, along with the patient’s age, overall well-being, and any additional medical conditions. 

Presently, the selection of treatment strategies for NSCLC is contingent upon both the disease stage and the overall health status of the patients. It should be noted that the results of the application of radiation therapy alone, in this case, remain unsatisfactory. According to prospective randomized studies, the median life expectancy is about 9–10 months, and the 5-year survival rate is 3–6%. The reason for the failures is the low sensitivity of NSCLC to radiation, as well as the fact that about 80% of patients with stage III NSCLC already have subclinical distant micro-metastases. In turn, the resistance to treatment is associated with a 1–5% population of cells—cancer stem (or stem-like) cells (CSCs). Active studies of the characteristics of CSCs, including the mechanisms of their increased resistance to ionizing radiation exposure, have recently begun. However, despite the large number of experimental and clinical studies, the exact mechanism of CSC resistance to ionizing radiation is still unclear [17,18].

Currently, there is no ideal radiation therapy (RT) method, and it causes damage to normal tissue, which limits its use in tumor treatment. The potential biological effects of IR dose to organs at risk and target volumes within the body are a dominant dose-limiting constraint in RT [19]. Reducing damage to surrounding healthy tissues has always been a focal point of interest in radiation therapy research. Current preclinical studies are showing positive results in sparing normal tissues, but gathering more systematic toxicity data is still necessary [20]. FLASH radiotherapy (FLASH-RT) delivering ultra-high dose rate (UHDR) has the potential to overcome the limitations of conventional RT (CONV-RT) [21]. Implementing intensity-modulated radiation therapy (IMRT) with UHDRs has reemerged as a promising treatment approach to effectively reduce potential damage to healthy tissue without compromising tumor control efficacy [22]. FLASH-RT is typically defined as a non-invasive external RT technology involving the delivery of UHDRs to the target volume at levels several orders of magnitude higher than those used in current clinical CONV-RT (≥40 Gy/s vs. 0.01–0.4 Gy/min, respectively) for an extremely short duration of time [23,24,25]. The overall time for dose delivery with FLASH-RT is significantly quicker (0.1 s) than DNA repair and other biological processes (minutes) but remains considerably slower than the radiolysis of water molecules through radio-chemical processes (10−16−10−7 s) [26]. CONV-RT affects the chemical and biological processes, whereas FLASH-RT has no impact on the biochemical steps (Figure 1). The DNA damage caused by FLASH-RT is lower than that caused by the CONV-RT dose rate [23,27,28,29]. In vitro studies have suggested that brief bursts of IR pulses lasting less than a millisecond induce fewer genetic abnormalities compared with continuous, prolonged exposure to the same cumulative radiation dose [25]. Research on FLASH-RT is important for confirming if this technology will change the way tumors are treated in the future. If clinical trials confirm the significantly improved safety and effectiveness of FLASH-RT, this cutting-edge technology could potentially revolutionize the field of radiation oncology, emerging as the primary modality for specific tumor types and potentially displacing CONV-RT in the future [30].

It is important to highlight the potential use of inducing free radicals during microwave exposure due to their therapeutic effect on tumor cells. Although the sensitizing effect of microwave irradiation in ionizing radiation of tumor cells was described in the pioneering work of F. Dietzel as early as 1975 [31], high-power microwaves (HPM) have only recently introduced innovative technologies and improvements in existing non-ionizing radiation approaches. In many instances, microwaves can also be used to effectively combat different types of cancers [32,33,34]. According to recent scientific studies, microwave radiation has been demonstrated to cause tumor cell apoptosis [35]. It was shown that W-band millimeter-wave (MMW) radiation, operated at a non-thermal power density of 0.2 mW/cm^2^, led to notable morphological changes associated with the apoptosis and senescence of human lung cancer H1299 cells [36]. The results were treatment-specific and depended on the amount of energy used, with no side effects expected after MMW radiation [36]. Microwave radiations induce cell apoptosis in lung cancer A549 cells by causing morphological changes and cell shrinkage [37]. A recent study also demonstrated that specific doses of HPM pulses induce apoptosis in human glioblastoma U87-MG cells [38]. The same research team studied how pulsed HPM affects NSCLC cell lines (H460 and A549) and compared them to normal lung MRC5 cells [39]. HPM primarily increases intracellular reactive species levels through a strong electric field of ∼27 kV/cm. Reactive oxygen species (ROS) play a crucial role in triggering HPM-induced cellular effects, with the possibility of NO species also contributing to their impact. Elevated DNA damage (upregulation of ATR/ATM, Chk1/Chk2, and p53 transcripts) and increased expression of apoptotic markers were observed. These alterations impact NSCLC viability, mitochondrial activity, and death rates observed during 72 h post-irradiation. While the effect on cancer cell viability in vitro and in vivo, along with acquired resistance to pulsed HPM, remains unexplored, this technique shows promise as a supplementary approach to non-surgical cancer treatments. It is worth mentioning that patients who receive microwave therapy before surgery exhibit a more favorable prognosis [40].

To date, uncomprehensive studies have been carried out on key processes responsible for the survival of CSCs after irradiation (Figure 2), in particular, (1) DNA repair processes; (2) the functioning of the p53-dependent system of preserving the stability of the genome and the activity of genes that affect the processes of proliferation and cell cycle control; (3) mechanisms of induction of programmed cell death; (4) influence of the composition of secreted EVs on EMT, including autocrine processes of self-renewal, migration of tumor cells and their radioresistance. According to the Clinicaltrials.gov database (accessed on 28 January 2024), there are currently 1056 clinical studies involving radiation therapy on NSCLC. However, only 368 studies have been completed, and results have been provided for 95 of them. DNA damage response as an outcome measure has been employed in two trials (NCT02221739, NCT00821951); the functioning of the p53-dependent system or influence of EVs in none; programmed cell death in one trial (NCT02434081). Published results from clinical studies show significant benefits of radiotherapy for NSCLC, but the common outcome measures are patient survival, tumor size, or metastases development. For example, in patients with advanced NSCLC, prior treatment with radiotherapy results in extended progression-free survival and overall survival when treated with pembrolizumab compared with those without prior radiotherapy [41]. The findings indicate a significant reduction in tumor size and metabolic activity following fractionated radiotherapy. Moreover, adjusting the radiation dose escalation to the fludeoxyglucose-avid tumor identified using midtreatment positron emission tomography yields favorable local-regional tumor control [42]. It was demonstrated that radiotherapy in patients with stage III locally advanced NSCLC without disease progression resulted in improved 5- and 10-year disease-free survival [43]. Similarly, chemotherapy of NSCLC followed by consolidative radiation therapy, without additional maintenance chemotherapy, demonstrates promising long-term outcomes [44]. 

## 3. Radioresistance and Radiosensitivity—Current Controversies

The primary approach for treating patients with NSCLC involves radiation therapy. Nonetheless, the resistance of tumor cells to radiation exposure stands as a prominent factor limiting the success of therapeutic outcomes. Radioresistance (RR) is intricately linked to processes such as epithelial-mesenchymal transition (EMT), cancer stem cell (CSC) characteristics, and oncogenic metabolism. Targeting these elements holds promise in enhancing the efficacy of radiotherapy, potentially preventing tumor recurrence and metastasis [2]. The induction of metastasis, the CSC phenotype, and oncogenic metabolism in cancer cells can be triggered by ionizing radiation (IR).

Multifractionated radiation (MFR) therapy serves as a primary approach for treating patients with NSCLC. Nonetheless, the resistance of cancer cells to radiation exposure remains the primary factor limiting successful therapeutic outcomes. To investigate the molecular and cellular mechanisms underlying the increased resistance of NSCLC to IR, two NSCLC cell lines were compared: A549 cells (with wild-type p53) and H1299 (p53 null) [45]. By subjecting these cells to MFR at a clinically relevant total X-ray irradiation dose of 60 Gy, survived populations were obtained, designated as A549IR and H1299IR. Further analysis revealed multiple alterations in these cells compared with parental NSCLC cells, including a more pronounced EMT-like phenotype observed in H1299IR (p53-deficient) cells compared with A549IR (p53 wild-type) cells. Remarkably opposite to parental cells, the clonogenic cell potential of the A549IR (p53 wild-type) significantly increased. In contrast, in the H1299IR (p53-deficient) cell, it was decreased, suggesting that functional p53 is essential for clonogenic survival of X-ray-resistant sublines (derived from MFR-exposed parental cells). A single acute exposure to a 2 Gy dose of X-rays led to a corresponding decrease of both γH2AX and pATM foci, the DNA DSB markers, to almost control level in A549IR and parental cells, thus suggesting repair of most of the DSBs within 24 h [5]. In contrast to parental cells, which demonstrated a sharp spike of both DNA DSB markers observed at the first-hour post-IR, the H1299IR cells exhibited a slight alteration in the number of γH2AX foci and a minimal change in the kinetic of pATM foci, with values nearly returning to background levels by the eighth-hour post-IR. The presence of functional p53 may be crucial for achieving the desired reduction in DNA-replicating cells and cells in the growth-pre-replicative phase in response to acute IR exposure, both in MFR-resistant and parental cells. Obtained data suggested that functional p53 sensitizes both parental and IR-resistant cells to apoptosis in response to single acute doses of IR. Moreover, it influences ABCG2 expression in cells that survived after fractionated IR exposure and points to a significant role of p53 in assigning a stem-like cell phenotype and RR to NSCLC cells that are associated with ABCG2 overexpression [46,47,48]. 

Conversely, the deficiency of p53 in parental cells results in a notable apoptotic response only at the highest dose of acute single-dose IR, leading to reduced apoptotic reaction in IR-resistant cells across all single acute doses. Moreover, there are indications that forceful and p53-independent recruitment of 53BP1 to damaged chromatin, along with 53BP1-mediated non-homologous end-joining (NHEJ), favors either the survival advantage of p53-null cells (exhibiting increased proliferation/DNA replication and decreased apoptosis) or the stimulation of an EMT-like phenotype [49]. It has been observed that distinct DNA repair mechanisms are activated by MFR and single-dose IR and that increased cell viability after MFR relies on both p53 and 53BP1 signaling pathways, along with NHEJ. Co-incubation of EVs isolated from parental H1299 and A549 cells and HUVECs induced proliferation (EdU) in endothelial cells more in the case of previous irradiation exposure on lung cancer cells in the dosage of 4 Gy, no difference was detected between proliferation activity of HUVECs themselves and after the addition of HEK239 derived EVs (control samples) [50]. Moreover, it was shown that EVs from NSCLC cells, which were not influenced by IR, promote apoptosis in BEAS-2B cells more than proliferation [51]. However, a single high dose of radiation (20 Gy) on xenograft models showed significant inhibition of tumor growth, abscopal effect and SASP phenotype due to the vesicular transport and p53-status of the cells (inhibition occurred in A549 xenografts only) [6]. 

There are discrepancies concerning mitochondrial transfer as the regulators of malignant cells’ RR. Indeed, it is highly likely that within tumor tissue, there may be a specific population of “Mt receivers” that could greatly impact their susceptibility to radiation. On the other hand, could dysfunctional mitochondria help cancer cells resist radiation damage? This dilemma led to the assumption that putting healthy cell mitochondria into radioresistant tumor cells makes them sensitive to radiation again. Consistent with this assumption, pure mitochondria isolated from normal human astrocytes and immediately co-incubated with starved human either radiosensitive (RS) (wild-type) or RR (rho0) glioma U87 cells could enhance the radiosensitivity of gliomas in vitro and in vivo [52]. The internalization of isolated mitochondria through endocytosis was driven by the intricate signaling cascade of NAD+-CD38-cADPR-Ca2+. Notably, the transfer of mitochondria through endocytosis was observed in approximately 33% of U87 cells, consistent with the 41% fraction of the MDA-MB-231cells that were the TNT-mediated “Mt receivers” in the co-culture with immune cells [53]. However, the transfer unexpectedly enhances the radiosensitivity of both RS (wild-type) and RR (rho0) glioma U87 cells, albeit at two-fold different extents. Mt endocytosis enhances gene and protein expression associated with the tricarboxylic acid (TCA) cycle, leading to increased aerobic respiration. It was previously shown that highly metastatic lung cancer cells are p53 defective, while non-metastatic cells have functional p53. Therefore, p53 is the first candidate as a potential biomarker for aggressive metastatic behavior of cancer cells and as a candidate for transfection. Additional genetic changes (K-RAS(V12), p53 knockdown, mutant EGFRs), commonly found in human lung cancer, progress normal human bronchial epithelial cells partially but not completely toward malignancy [54,55]. Therefore, novel models are highly required to track and monitor changes in malignant phenotype, radiosensitivity (RS), colony formation and trans-differentiation in response to the transplantation of mitochondrial cargo of MtO and MtS/MtO-EVs isolated from RR and RS NSCLC cells. Moreover, the mitochondria isolated from these and/or human mesenchymal stem cells can be tested as a therapeutic tool to modulate the RR of NSCLC cells (Figure 3).

## 4. Extracellular Vesicles (EVs)

In recent times, considerable focus has been directed towards investigating the role of extracellular vesicles (EVs), exosomes in particular, secreted by tumor cells as molecular messengers and prognostic biomarkers [56]. EVs, which are carriers of molecular information, play an important role both in the autocrine regulation of homeostasis of CSCs themselves and in paracrine maintenance of dynamic equilibrium and signaling between the CSC population and stromal cells of the tumor microenvironment. MicroRNAs transported by EVs are involved in the regulation of EMT processes, self-renewal and differentiation of CSCs, and carcinogenesis, as well as in the reaction of cells to anti-cancer therapy. It is well-known that the repair of DNA double-strand breaks (DSBs) induced by ionizing radiation in the CSC population proceeds more efficiently [18]. In this regard, it is very important to investigate the spectrum of exosomal miRNAs secreted by irradiated cancer cells and their role in the regulation of the increased expression of proteins involved in both EMT and the repair of double-strand breaks in DNA [57,58]. 

During the cell cycle, all cells release EVs, such as exosomes, which contain proteins, and mRNAs and DNAs capable of influencing proliferation and triggering apoptosis in recipient cells. EVs emitted by metastatic cells play a crucial role in promoting tumor cell invasion and preparing metastatic niches. Mounting evidence proposes that EVs can transfer genetic material to recipient cells. While it is expected that the proteome of the exosomes mirrors that of the originating cell, the protein cargo of exosomes from cancer cells can undergo modifications [59]. However, the mechanism and significance of this phenomenon remain largely unknown. Queries accompanying the function of exosomes predominantly focus on comprehending the destiny of their components and the phenotypic and molecular changes they trigger in recipient cells within cell-culture systems [59]. It was previously shown that exosomes secreted by cancer cells would sufficiently promote metastasis in various ways. For example, exosomes obtained from breast cancer and prostate cancer cells prompt neoplasia by transferring their miRNA cargo [60,61]. Additionally, the plasticity observed in cancer cells may, in part, be attributed to exosomes. Notably, exosomal miR-200 from metastatic breast cancer cells enhances the EMT and metastasis in breast cancer cells that are otherwise weakly metastatic [62]. 

Increasing evidence indicated that the EVs wheel the vicious cycle of autocrine/paracrine interaction of tumor cells and surrounding tissue cells, playing an essential role in both the metabolism of recipient cells and pre-metastatic niche formation (Figure 4). The indication that EVs carrying exosome-specific signatures secreted by metastatic NSCLC (both RR and RS) (carrying either p53 null or p53 wild-type, respectively) cells significantly increased the metabolic and proliferation activity of human embryonic lung fibroblasts with accompanying induction of certain oncogenes. In contrast, the EVs from RS NSCLC cells induced subtle responses independent of p53 functionality [5]. Interestingly, the exhausting of the EV population with NSCLC-specific marker sorbent leads to the complete elimination of the proliferation effect of RR NSCLC (p53 null) cells but not RR NSCLC (p53 wt) cell-derived EVs. These data suggest that the EV cargo has a p53-related context. Alternatively, p53 might code the different molecular content of EVs capable of inducing metabolic and proliferative effects on human lung fibroblasts [45].

Different cargo carried by cancer cell-derived exosomes, including nucleic acids, signaling proteins, and metabolites, can elicit pro-tumorigenic effects on cells (Figure 5). For instance, exosomes derived from breast cancer contain miR-122, which inhibits pyruvate kinase and, consequently, glucose uptake in the lungs, thereby promoting metastasis [63]. Conversely, exosomes derived from fibroblasts stimulate the migration of breast cancer cells by inducing planar cell polarity autocrine signaling [64]. Exosomes released by cancer cells were reported to promote chemo- and radioresistance in various types of cancer. Upon treatment of tumors with radiation therapy or a gamma-secretase inhibitor, the emergence of tumor-initiating cells resistant to radiation therapy occurs due to the transfer of CAF-derived exosomal RNA and transposable elements to cancer cells [65]. Furthermore, HER2-positive exosomes from breast cancer cells function as decoys for anti-HER2 therapy [66]. Also, chemo- and radiotherapy can directly influence exosome secretion and their content, potentially impacting therapy outcomes [67]. In opposition to their tumorigenic function and the interruption of cancer therapy, exosomes were proposed to be used in clinical applications as diagnostic or therapeutic agents. Specific miRNAs, or clusters of miRNAs present in exosomes, hold the potential for diagnostic or prognostic purposes in cancer detection [68]. Oncogenic and tumor-suppressor miRNAs in exosomes may offer significant diagnostic value due to their distinct expression patterns between cancer cells and normal cells [59]. Exosomes derived from mesenchymal stem cells (MSCs) have been proposed as a therapeutic modality on their own [69] or for facilitating the delivery of miR-146b and targeting EGFR in glioma in rat models [70]. Moreover, the administration of MSC-derived exosomes in the treatment of a patient with graft-versus-host disease demonstrated that repeated injections were well-tolerated, did not result in significant side effects, and led to a positive patient response [71].

## 5. Mitochondria in Cancer Development and Progression—Extraction and Functional Characterization

Mitochondria play crucial roles in cancer development and progression, influencing various cellular processes, such as energy metabolism, apoptosis, and reactive oxygen species (ROS) generation. Understanding the extraction and functional characterization of mitochondria in cancer is essential for unraveling their specific contributions to tumorigenesis and identifying potential therapeutic targets. Functional characterization of mitochondria in different cancer contexts aids in identifying common pathways and specific vulnerabilities that can be targeted for therapeutic intervention.

Mitochondrial defects amplify the RR of HepG2 cells in human liver cancer, potentially mitigating reactive oxygen species-induced oxidative damage and inhibiting the mitochondrial apoptosis pathway [72]. Additionally, the severity of mitochondrial structural abnormalities directly correlates with the degree of malignancy in breast tumors [73]. Mitochondrial structure abnormality and defects were found to be in high correlation with malignancy and RR [74]. Several mitochondrial proteins were shown to directly contribute to cancer invasiveness and may serve as potential biomarkers. For example, two subunits of ATP synthase (α-subunit and d-subunit) are overexpressed in cancers with high histological grade. The increased abundance of mitochondria-bound protein COX5A was demonstrated to play an active role in the migration and invasion of NSCLC cells. Single-stranded DNA binding protein (SSBP) regulates mitochondrial function and metabolism, and its level correlates with cancer aggressiveness [75].

Tracking whether the transfer of MtO and mtDNA from cancer cells occurs can help the understanding of the mechanism of cancer dissemination (metastases). Hence, a method for the preservation of the complete function of MtO extracted from cells is required. The current techniques for cell disruption to extract MtOs can be broadly categorized into nonmechanical and mechanical methods, each with its own set of advantages and drawbacks. Nonmechanical methods typically include osmotic shock, freezing and thawing, enzyme lysis, chemical treatment, and detergents [76,77,78]. However, these methods are prone to damaging the activity of extracted MtO due to repeated swelling and exposure to chemical substances. On the other hand, mechanical methods encompass homogenization, shaking with glass beads, fine grinding, and ultrasonication [79,80]. While mechanical methods generally offer better fragmentation efficiency compared with nonmechanical methods, the high pressure applied during these processes can potentially compromise MtO activity due to significant variations in pressure and temperature [81,82]. A recent development involves a centrifugal device designed for the efficient extraction of functional MtO using a centrifuge commonly found in standard laboratory settings [8]. When compared to the quantity and effectiveness of MtO isolated from an equivalent number of cells using other methods, MtO extracted using the device exhibits a more comprehensive mitochondrial electron transport chain complex and a comparable number of MtO. Moreover, transmission electron microscopy (TEM) revealed a typical MtO structure in device-extracted samples. Furthermore, the MtO membrane potential, as indicated by tetramethylrhodamine ethyl ester staining, was higher in device-extracted MtO than in kit-extracted counterparts [83]. 

Furthermore, a diagnostic system for the effective detection of the specific MtS (proteins, mitochondrial mRNA, mtDNA, etc.), which are feasible markers for malignancy identification, is also highly desired. Presently, ELISA and Q-PCR are used for the accurate detection of protein and nucleic acid, respectively. However, these methods are time-consuming, relatively costly, and necessitate highly trained personnel. One of the long-term ways of developing diagnostic systems is the development of biosensors and bioanalytical systems. The biosensor systems containing biomacromolecules, cells, and cellular organelles as “learning” elements are capable of providing the highly sensitive and fast analysis of chemical compounds and microbiological objects with the analytical and operational characteristics unattainable at the use of usual chemical sensors and physical systems of registration. In a biosensor, the sensitive element containing biological material (enzymes, tissues, bacteria, yeast, anti-genes/antibody, liposomes, organelles, receptors, DNA) directly reacting to the presence of the defined component generates a signal that is functionally connected with the concentration of this component [84,85]. This class of devices will allow it to pass to new extreme sizes and functionality, and it is also of fundamental physical interest in connection with an opportunity to operate with a power range of separate molecular elements. A highly sensitivity biosensor based on an electrode with a self-assembled monolayer of gold nanoparticles on a microhemisphere array has been recently developed [86]. This kind of biosensor may be used as to detect the specific MtS, such as proteins, mitochondrial mRNA, and mtDNA, hence diagnosing the cancer cell RR and dissemination. 

Mitochondrial content, especially damaged or oxidized components resulting from IR exposure, can act as damage-associated molecular patterns (DAMPs) capable of activating an inflammatory response either in the cytosol or upon release from cells [87]. The mitochondrial antiviral signaling protein (MAVS), situated in the mitochondrial outer membrane, serves as an innate immune signaling molecule implicated in the radiation response through its oligomerization mediated by radiation-induced reactive oxygen species (ROS). MAVS plays a role in the expression of IFN-β and IFN-stimulated genes in response to IR. MAVS complexes facilitate the nuclear translocation of the NF-κB and IRF3/7 transcription factors, consequently inducing the innate antiviral response [88].

EVs, as well as a circulating functional MtO and MtS, may be found in culture media and in any bodily fluid, including blood, urine, and saliva. However, the isolation of high-yield and purity EVs from body fluids still remains a challenge for productive research and development of methods for disease diagnostics and treatment. Various methods have been employed for EVs, MtO, and MtS extraction and characterization (Table 1) due to their high importance in the study of cancer radioresistance and therapy.

Recent advancements in mitochondria extraction and functional characterization techniques have greatly improved our understanding of their roles in cancer biology. These techniques have been applied to various types of cancer, resulting in significant contributions to our knowledge in this field. Research in this area has mainly focused on common cancer types like breast, prostate, and colorectal cancer. The methods used can also be applied to study mitochondrial dynamics and function in non-small cell lung cancer (NSCLC). Researchers can use advanced techniques to better understand the specific changes in mitochondrial biology that lead to the development, progression, and response to therapy of NSCLC.

## 6. Mitochondria Transfer between Cells

The transfer of mitochondria between cells, known as mitochondrial transfer, is a phenomenon observed in various types of cancer, including non-small cell lung cancer (NSCLC). There can be differences between cancer types, influenced by factors such as cellular microenvironment, genetic background, and specific characteristics of the cancer cells involved. Limited scientific publications exist on mitochondria transfer between NSCLC cells, hindering our understanding of its role in tumor progression and therapeutic potential. However, similar mitochondrial alterations, including increased oxidative phosphorylation and mitochondrial biogenesis, are observed across various cancer types, highlighting the broad relevance of mitochondrial dysfunction in cancer progression. 

Tunneling membrane nanotubes, macropinocytosis and endocytosis have been suggested as the most probable pathways for mitochondrial transfer between cells (Figure 6). Nevertheless, the precise physiological nature and specific mechanism of mitochondrial transfer remains vague [52]. In addition, the question of how and in which way mitochondrial information can confer IR-induced EMT/CSC/oncogenic metabolism and may modulate resistance to radiotherapy remains to be investigated.

Various techniques have been employed for mitochondria transfer research (Table 2). The impact of mitochondria transfer in cancer is ambiguous. The data showcase increases in proliferation and invasiveness after the addition of MSCs’ mitochondria to the MDA-MB-231 breast cancer cell line. However, cells treated beforehand with cisplatin and rodamin red 6 (reagent for mitochondria dysfunction) displayed enhancement of cisplatin-induced apoptosis, chemosensitivity, and decrease in migration [106]. Mitochondria transfer plays a vital role in tissue renewal, and recent studies have displayed promising outcomes of such therapy in pulmonary fibrosis, where MSCs are the source of mitochondria [107]. Mitochondrial functionality is indispensable for CNS disorders recovery. Chemotherapy with cisplatin induces cognitive deficits that could be treated within mitochondria transfer from mesenchymal stem cells or astrocytes [108]. It has been shown that the transfer of mitochondria enhances the proliferation, migration, and osteogenic differentiation of bone marrow mesenchymal stem cells, contributing to the promotion of bone defect healing [109]. The findings suggest that mitochondrial transfer could represent a novel and promising approach for enhancing the therapeutic efficacy of stem cells.

The horizontal transfer of mitochondria transpires via the conveyance of either mitochondria-derived vesicles or intact mitochondria, facilitated by transformation, conjugation, or tunneling nanotubes (Table 3). Currently, the precise mechanisms governing the intercellular transfer of free mtDNA across both mitochondrial inner and outer membranes, as well as the plasma membrane, are still not fully understood. Moreover, the transfer of entire mitochondrial particles represents the most probable scenario facilitating the restoration of mitochondrial function through EVs during intercellular mitochondrial transfer. In this process, certain cells acquire a few mitochondria and replicate their mtDNA [117]. EVs have the capability to carry the complete mitochondrial genome. Subsequently, these EVs can transfer their mitochondria and/or mtDNA to cells exhibiting impaired metabolism, thereby facilitating the restoration of metabolic activity. The horizontal transfer of mitochondria and mtDNA itself has been observed in cancer stem-like cells, correlating with heightened self-renewal potential and resistance to chemotherapy and radiotherapy. It may be hypothesized that mitochondria and/or mitochondrial signature (MtS - e.g., mitochondrial proteins, ligands, mtDNA, etc.) transfer occurs in NSCLC via EVs and these EVs are essential for the maintenance of RR of cancer cells.

Through various experimental approaches, researchers have uncovered mechanisms by which mitochondria can be exchanged between cells, including tunneling nanotubes, extracellular vesicles, cell fusion, and the intercellular transfer of organelles via membrane nanotubes. These approaches have helped us understand how mitochondria are delivered in various biological situations and in different types of cells. The research has focused on understanding how mitochondria are delivered in different cell types, and these findings can be used to study mitochondrial delivery in non-small cell lung cancer (NSCLC). Exploring the potential role of mitochondrial exchange in the progression, metastasis, and resistance to treatment of NSCLC can be achieved by using these common experimental approaches.

## 7. Conclusions and Future Directions

The cytotoxic effect of radiation on cancer cells relies on mitochondria, and the transplantation of MtO, MtDNA, or proteins could potentially serve as an effective strategy for enhancing radiosensitivity (Figure 3) [52]. A novel teranostic approach against malignancy based on the identification and characterization of MtO and/or MtS/MtO-EVs carrying metabolic code should be established with the prognostic significance in conferring RR and metastatic potential dictated by NSCLC. The investigation of the functional connections between DNA repair and inflammation/immunity pathways that are physiologically affected during IR and contribute to post-irradiation cancer progression is highly important [123]. The clarification of the cross-talk between these cellular processes and their combined effects in RR and stemness will significantly extend our understanding of the mechanisms underlying the IR-related loss of stem cell function and will pave the way to innovative MtO and/or MtS/MtO-EV-based interventions to improve the lifespan and healthspan of the patients.

New information on the influence of the CONV and FLASH irradiation regimen (power and dose fractionation) on the survival rate of RR and CSC-like cells should be obtained [30]. The effect of the pharmacological modulators of EV secretion, DNA repair, and autophagy on the resistance of NSCLC cells to irradiation should be further investigated. Understanding the role of Mt and EV signatures and their metabolic reprogramming code is a very important task [7,124]. As a result, the achievement of this task will lead to exploring the teranostic approach, including (i) developing the biosensor chip-based analysis of circulating MtO and MtS/MtO-EVs as a diagnostic and prognostic tool and monitoring the success of anti-cancer treatment; (ii) testing the MtO and MtS/MtO-EVs with edited metabolic code as a therapeutic tool to reduce metastatic potential and to augment the sensitivity of NSCLC and CSCs to radiotherapy. 

Comprehending the molecular mechanisms underlying RR and the metastatic capabilities of malignant tumors, especially those that may be reversible, is crucial for developing therapies capable of attenuating or halting cancer progression in human tissues. Targeting the specific causes or bolstering protective processes to sustain healthy function is vital in this regard. Despite rapid advances and the excitement regarding the combination of mitochondria-driven therapy and radiotherapy (in both academic and clinical arenas), several main obstacles remain if these approaches are to reach their full potential: deeper, integrated understanding of determinants of cancer cell RR and dissemination; developing improved natural methods for mitochondria-based clinical applications; understanding the effects of the local microenvironment on cancer cell RR and its importance during radio- and chemotherapies; establishment safe and effective therapeutic and diagnostic tools to overcome RR and to monitor the therapeutic efficacy and prognosis. 

Summarizing the available literature and experimental findings on the transfer and transplantation of mitochondria between cells across various cancer types, the most probable mechanisms facilitating this transfer include tunneling membrane nanotubes, macropinocytosis, and endocytosis, which can be accomplished through various techniques such as co-incubation, microinjection, or transplantation. Mitochondria transfer’s impact on cancer is significant [10]. Various research techniques have been developed and used for different types of cancer, such as breast, kidney, skin, bone, bladder, brain, and lung [76,77,78,79,80,81,82,83,84,85,86,110,111,112,113,114,115,116]. The techniques used to extract and study MtO and EVs, as well as their implications for mitochondria transfer-related DNA damage response, can also be explored in NSCLC. The ongoing investigations into radiation therapy methods such as FLASH-RT and HPM techniques show great potential to revolutionize the treatment of NSCLC. Recent progress in preclinical research and the possible insights gained emphasize the urgent need for thorough toxicity assessments and robust clinical trials. A sufficient number of ongoing clinical studies on the relationship between mitochondria (12 studies according to Clinicaltrials.gov database, accessed on 10 May 2024) or EVs (11 studies according to Clinicaltrials.gov database, accessed on 10 May 2024) and radiation therapy for cancers like breast, prostate, pancreas, and skin reflects a growing interest in refining treatment approaches and enhancing outcomes. These trials also open up new perspectives that may redefine the standards of care for NSCLC. 

Future research in the field of extracellular vesicles (EVs) holds promise for advancing our understanding of their role in non-small cell lung cancer (NSCLC) and improving therapeutic strategies [3,4,5,6]. Investigating the molecular mechanisms underlying EV-mediated intercellular communication and mitochondrial transfer in NSCLC is paramount. Additionally, there is a need to explore the potential prognostic significance of EV cargo, such as mitochondrial organelles (MtO) and their proteinaceous cargo signatures (MtS), in predicting radioresistance and metastatic potential in NSCLC patients. Developing innovative teranostic approaches based on the identification and characterization of EVs carrying metabolic code may provide valuable diagnostic and prognostic tools for NSCLC. Furthermore, targeting EV secretion pathways and elucidating their roles in DNA repair and autophagy could lead to novel therapeutic interventions aimed at overcoming radioresistance in NSCLC cells. Overall, future research directions should focus on unraveling the complex biology of EVs in NSCLC and translating these findings into clinical applications to improve patient outcomes.

The role of mitochondria in non-small cell lung cancer (NSCLC) radioresistance and radiosensitivity presents exciting avenues for future research. Investigating the functional connections between DNA repair, inflammation, and immunity pathways affected during irradiation is crucial for understanding post-irradiation cancer progression and developing targeted interventions [124,125,126]. Moreover, elucidating the molecular mechanisms underlying mitochondrial DNA damage response and its implications for NSCLC therapy is essential. Future studies should explore the potential of pharmacological modulators targeting mitochondrial function, DNA repair, and autophagy to enhance the sensitivity of NSCLC cells to radiation therapy. Additionally, developing safe and effective therapeutic and diagnostic tools, such as biosensor chip-based analysis of circulating mitochondrial organelles (MtO) and their cargo, may revolutionize NSCLC treatment strategies. Collaborative efforts between researchers and clinicians are needed to translate preclinical findings into robust clinical trials, ultimately improving treatment outcomes for NSCLC patients. Overall, future directions in mitochondrial research aim to harness the therapeutic potential of mitochondria-based interventions and enhance our understanding of their role in NSCLC radioresistance and radiosensitivity.

The intricate interplay between extracellular vesicles (EVs) and mitochondria plays a crucial role in modulating the radioresistance and radiosensitivity of non-small cell lung cancer (NSCLC). EVs facilitate the transfer of mitochondria organelles (MtO), mitochondrial DNA (mtDNA), and/or mtRNA/proteinaceous cargo signatures (MtS) between cells, potentially influencing cellular responses to radiation therapy. These EV-mediated transfers of mitochondrial components may contribute to the restoration of mitochondrial function in NSCLC cells, thereby enhancing their resistance to radiation. Conversely, dysfunctional mitochondria or mitochondrial DNA mutations, when transferred via EVs, may sensitize NSCLC cells to radiation by impairing cellular energetics and increasing oxidative stress. Thus, the dynamic interaction between EVs and mitochondria in NSCLC highlights the complexity of cancer biology and emphasizes the necessity for additional research to understand the underlying mechanisms and to target therapeutic opportunities that can overcome radioresistance and enhance radiosensitivity in NSCLC.

The controversy surrounding the role of EVs and mitochondria in modulating the radioresistance and radiosensitivity of NSCLC underscores the complexity of cancer biology and the challenges in therapeutic development. While some studies suggest that alterations in mitochondrial function contribute to radioresistance by influencing cellular pathways involved in DNA repair and apoptosis, others propose that dysfunctional mitochondria or mitochondrial DNA mutations may actually sensitize NSCLC cells to radiation by impairing cellular energetics and increasing oxidative stress [125,126]. These contradictory findings underscore the complexity of mitochondrial biology and its interactions with radiation response pathways in cancer cells. The identification of novel therapeutic targets presents an opportunity to develop alternative strategies aimed at hindering the growth, drug resistance, and metastasis of NSCLC by modulating their metabolic characteristics. The expectation is to uncover specific metabolic inhibitors capable of overcoming the characteristics of NSCLC, thereby hindering tumor recurrence and metastasis and potentially leading to a lasting cure for cancer patients. Disrupting the adaptable capabilities of NSCLC through metabolic signaling blockade may unveil numerous avenues for discovering novel therapeutic targets suitable for personalized cancer treatments in clinical settings.

## Figures and Tables

**Figure 1 cancers-16-02235-f001:**
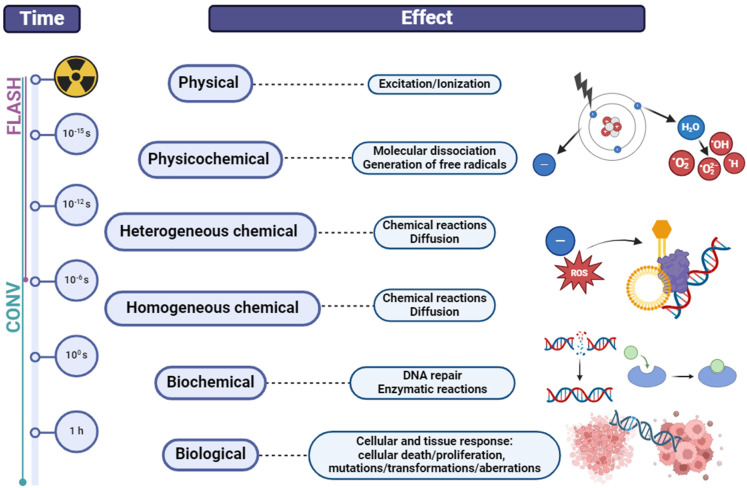
Timescales and effects of ionizing radiation from the initial physical excitation and ionization events occurring in femtoseconds to subsequent physicochemical effects like free radical generation over picoseconds, chemical reactions over nanoseconds to seconds, biochemical processes like DNA repair over seconds to minutes, and ultimately biological effects at the cellular and tissue level from mutations and transformations taking hours. Created with BioRender.com (accessed on 10 April 2024).

**Figure 2 cancers-16-02235-f002:**
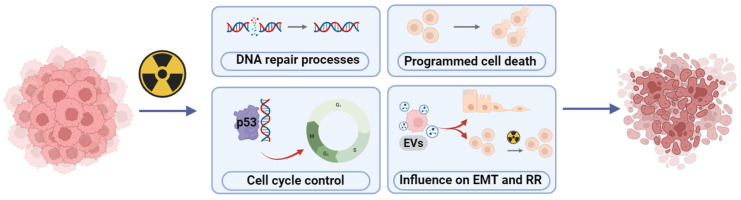
Involvement of irradiated CSCs in key cellular mechanisms and molecular pathways such as DNA repair processes, programmed cell death, cell cycle control by the p53-dependent system and EVs impact on EMT and RR that are responsible for their survival. Created with BioRender.com (accessed on 28 January 2024).

**Figure 3 cancers-16-02235-f003:**
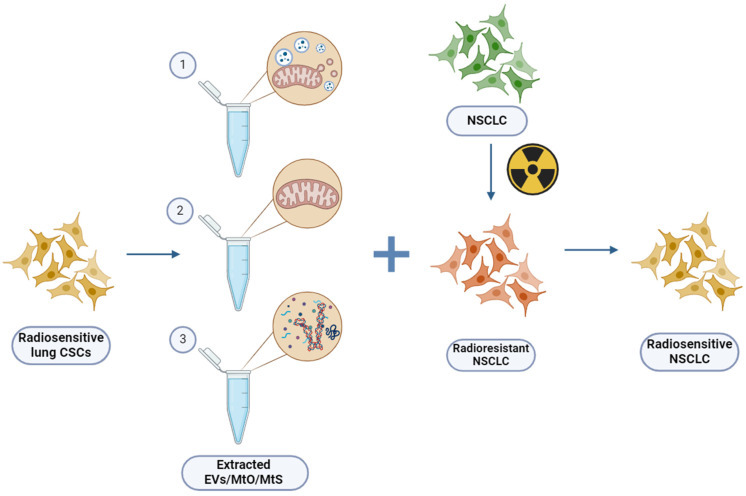
Treatment of radioresistant (RR) non-small cell lung cancer (NSCLC) cells by EVs/MtO/MtS (1/2/3 respectively) extracted from radiosensitive (RS) cancer stem (or stem-like) cells (CSCs) will lead to their radiosensitivity or decrease in RR, and thus will improve treatment protocols and increase the survival rate in NSCLC. Created with BioRender.com (accessed on 28 January 2024).

**Figure 4 cancers-16-02235-f004:**
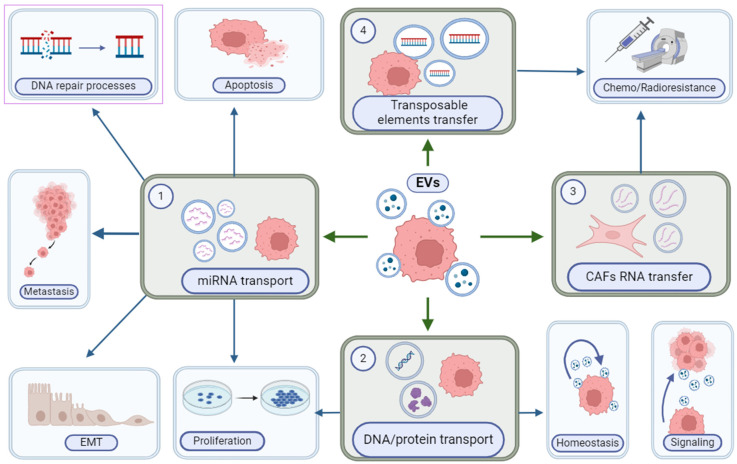
The main functions of extracellular vesicles (EVs) in cancer depend on the cargo inside them. (1) EVs containing miRNA trigger metastasis and epithelial-mesenchymal transition, increase proliferation activity and display a positive effect on DNA repair processes; some miRNA types activate apoptosis, which can be a useful tool for novel therapeutic strategies. (2) DNA/protein transport has a significant influence on signaling processes between cancer cells and the tumor microenvironment, maintains cell homeostasis, and facilitates proliferative function. (3) EVs received from CAFs with RNA, and (4) Exosomes with transposable elements promote chemo- and radioresistance. Created with BioRender.com (accessed on 28 January 2024).

**Figure 5 cancers-16-02235-f005:**
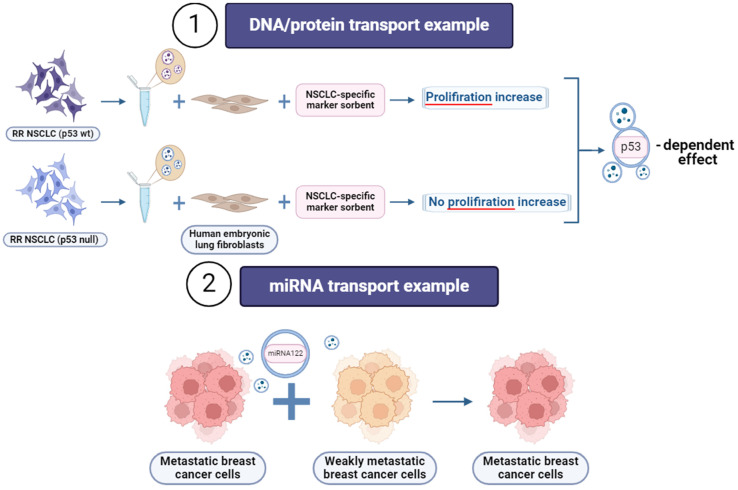
Examples of vesicular transport effect in cancer. (1) Identification of the role of EVs transporting p53 motif (DNA/protein transport type of vesicular functions in cancer) from RR NSCLC in an increase of human embryonic lung fibroblasts proliferation rate by the comparison of the effect of the vesicles from two cell phenotypes—p53 containing and p53 null in the state of EVs depletion by specific sorbent. (2) Signaling between more and less metastatic breast cancer cells reverses the second one to a more invasive phenotype by miRNA transport (miRNA type of vesicular functions in cancer). Created with BioRender.com (accessed on 28 January 2024).

**Figure 6 cancers-16-02235-f006:**
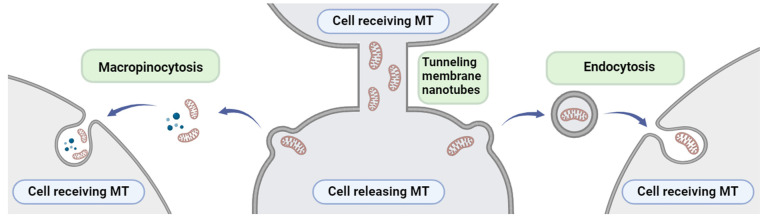
The common pathways for mitochondrial transfer between cells. Created with BioRender.com (accessed on 28 January 2024).

**Table 1 cancers-16-02235-t001:** Extracellular vesicles (EVs), mitochondrial organelle (MtO), mitochondrial signature (MtS-e.g., mitochondrial proteins, ligands, mtDNA, etc.) common extraction and functional characterization methods.

Organelle	Extraction Method	Functional Characterization	Ref.
**EVs**	Precipitation	Western blotting	[89,90]
	Differential ultracentrifugation	Nanoparticle tracking analysis	[91,92]
	Size exclusion chromatography	Electron microscopy imaging	[93,94]
**MtO**	Differential ultracentrifugation	Biuret methods	[76,95]
		Western blotting	[96,97]
		Imaging	[98,99]
**MtS**	Differential ultracentrifugation	Mass spectrometry	[100,101]
	Alkaline extraction	Multiplex PCR	[102,103]
	Precipitation	SYBR green-based PCR	[104,105]

**Table 2 cancers-16-02235-t002:** Common methods for mitochondria transfer and transport based on different experimental approaches employed to facilitate the transfer of mitochondria between various cell types.

Technique	Description	Result	Ref.
**Co-incubation**	Isolated mitochondria from healthy fibroblasts and mitochondrial-mutated cells were transferred into breast cancer (MCF7), embryonic kidney (HEK 293), and hepatocellular carcinoma (HepG2) cells	Mimic natural mitochondria transfer and persistent in the recipient cells for several days	[110]
	Following apoptosis induction, CellTracker-labelled pheochromocytoma (PC12) cells were incubated with the conditioned medium from untreated and unlabeled PC12 cells	Rescue effect is nulled because there was no mitochondria transfer, indicating a contact-dependent mechanism	[111]
	Incubation of metastatic breast cancer cells (MCF7) with extracellular vesicles from murine cancer-associated fibroblasts (CAFs)	Transfer of therapy resistance to therapy-sensitive cells via mtDNA from EV *in vivo* and *in vitro*	[112]
**Microinjection**	Mouse melanoma (B16ρ^0^) cells transfected with a plasmid coding for nuclear-targeted blue fluorescent protein (nBFP) were injected subcutaneously into C57BL/6N*^su9-DsRed2^* mice (transgenic mice expressing red fluorescent protein in somatic cell mitochondria (the *CAG/su9-DsRed2-*transgene)	Double-positive cells with both red and blue fluorescence, prepared from a pre-tumor lesion, identifying mouse stromal cells as a source of mitochondria	[113]
	Mitochondrial transfer from bone marrow-derived stromal cells (BMSCs) to primary human acute myeloid leukemia (AML) cells injected into immunodeficient NSG mice via acute myeloid leukemia AML-derived tunneling nanotube (TNT)	The transfer was enhanced by the treatment with hydrogen peroxide that drives the spike in ROS level in BMSC	[114]
	Injection of mCAF extracellular vesicles into tumor-bearing mice	Transfer of therapy resistance to therapy-sensitive cells via mtDNA from EV *in vivo* and *in vitro*	[112]
**Transplantation**	Mitochondria were transported into MCF-7 breast cancer cells through passive uptake or peptide Pep-1-mediated delivery	Mitochondria and peptide showed significant induction of the nuclear translocation of apoptosis-inducing factor	[115]
	In vitro mitochondrial transfer among bone marrow-derived mesenchymal stem cells (BM-MSCs) and two additional populations of MSCs sourced from healthy lung tissues (LT-MSCs) and bronchoalveolar lavage fluid of lung transplant recipients (BAL-MSCs)	LT-MSCs and BAL-MSCs exhibit the ability to donate spontaneously cytoplasmic content and mitochondria to healthy human bronchial epithelial cells with comparable efficiency through unidirectional transfer	[108]
	H9c2 rat heart myoblast cells and L929 mouse fibroblast cells were treated with uncoated or fluorescently coated mitochondria obtained from HeLa cells	Uptake and intracellular localization of HeLa-derived mitochondria in H9c2 cardiac myoblast cells were recorded	[116]

**Table 3 cancers-16-02235-t003:** Results of horizontal MtO/MtS transfer.

Method	Cells	Result	Evidence	Ref.
**Transformation**	B16ρ^0^SC B16ρ^0^CTC B16ρ^0^SCL	MtDNA is transferred from stromal cells to B16ρ^0^ cells within intact mitochondria	Acquisition of mtDNA by the trafficking of whole mitochondria from host donor cells to ρ^0^ cells resulting in long-lasting respiration recovery and efficient tumor formation	[113]
**Conjugation**	IMR90 WI-38 MDA-MB-157 U2OS A382 HCC1806	The transfer of mtDNA most likely occurs through either the transfer of mitochondria-derived vesicles or intact mitochondrial organelles	Identification of variants exclusive to the non-GFP-labeled cell line within the co-cultured partner cell line indicates the transfer of mtDNA between the cells	[118]
**Tunneling nanotubes (TNTs)**	T24 RT4	The distribution of mitochondria transferred from T24 cells was in good agreement with the original mitochondria in RT4 cells, which may indicate mitochondrial fusion	The indication that TNTs promote intercellular mitochondrial organelles transfer between heterogeneous cells and the transfer is unidirectional	[119]
**Tunneling nanotubes (TNTs)**	4T1 4T1p^0^	The displayed horizontal transfer of mtDNA from normal host cells to tumor cells lacking mtDNA was clearly established	The mtDNA transfer results in recovery of respiration, tumor initiation and metastasis	[120]
**Tunneling nanotubes (TNTs)**	Primary MM MM1S U266	Increased level of ATP and oxidative phosphorylation in MM cells	CD38 is required for the formation of TNTs facilitating tumor mitochondrial transfer	[121]
**Tunneling nanotubes (TNTs)**	PC12	Increased death rate of UV-treated cells co-cultured with *ρ*^0^ cells, compared with cells carrying functional mitochondria; indication of mitochondria transferred from untreated cells	Successful mitochondria transfer displayed its participation in the rescue effect by preventing apoptosis in its early stage in damaged cells, which form a novel type of TNTs	[111]
**Extracellular vesicle (EV)**	RAS-3	Extracellular vesicles mediate intercellular transfer of oncogenic human H-ras DNA	The indication of an avid uptake of EVs	[122]

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
