# Peer review of "Extracellular Vesicle- and Mitochondria-Based Targeting of Non-Small Cell Lung Cancer Response to Radiation: Challenges and Perspectives"

_cancers, 2024, doi:10.3390/cancers16122235_

Round 1

Reviewer 1 Report

Comments and Suggestions for Authors

The topic may be interesting, but this review is written in a language that is difficult to follow.

The sections are not well organised: The authors need to introduce the topic and then develop the scientific data with more clarity.

At the end of section 2, the authors need to discuss the outcome and the limits of knowledge about the cellular and molecular mechanisms of radioresistance.

Sections 4 and 5, which are central to the topic, contain a wealth of information about cells that are different from those of lung cancer. This is one of the major inconsistencies because there are no consistent scientific reports in the manuscript on NSCLC. On the contrary, there are described other types of cancer cells and this seems too speculative.

The controversy on this topic (as cited in the abstract) has not been justified and discussed.

Minor

The figures are not mentioned in the text. In addition, Figure 3 in the figure legend describes different processes, which are numbered. However, the numbers corresponding to the processes are not shown in the figure.

Line 57: Please clarify the acronyms MtO and MtS/MtO-EVs and RR

Line 167: “in the regulation of expression increased expression of proteins” Something is wrong with this sentence...

Lines 148-154: I think these considerations should be moved to the Conclusion section.

Author Response

Thank you for your feedback on our manuscript. We appreciate your insights and suggestions for improvement.

We acknowledge your concerns regarding the organization of the manuscript. We work diligently to enhance the clarity of the language and ensure that the sections are well-organized, as detailed in the attached file.

Thank you very much again for your valuable feedback. We are committed to addressing your concerns and improving the manuscript accordingly.

Reviewer 2 Report

Comments and Suggestions for Authors

Radiation therapy is a key treatment for non-small cell lung cancer (NSCLC), but resistance remains a challenge. Mitochondrial abnormalities correlate with malignancy and radioresistance. Mitochondrial transplantation could enhance radiation sensitivity. Extracellular vesicles (EVs) transfer genetic material, including mitochondria, influencing cell behavior. This transfer affects cancer radioresistance and sensitivity. The review aims to elucidate the mechanisms behind these findings. The manuscript required some modification before publication.

1. For a comprehensive review article, I recommend enhancing the introduction section by providing more background information pertinent to NSCLC.

2. Highlight the novelty of this review in the abstract section.

3. I suggest including an explanation of the role of reactive species post-irradiation in the review. I believe the induction of reactive species post-exposure played a key role in inducing cellular effects.

4. I suggest incorporating a discussion on the cellular effects observed following the irradiation of nanosecond pulsed high-power microwave on NSCLC, as outlined in a recent study (https://doi.org/10.1016/j.fmre.2024.02.001), into this review.

5. I recommend thoroughly reviewing the manuscript to identify and rectify any typos and grammatical errors

Comments on the Quality of English Language

I recommend thoroughly reviewing the manuscript to identify and rectify any typos and grammatical errors

Author Response

We sincerely appreciate your thorough review of our manuscript. We are delighted that you found our exploration of mitochondrial abnormalities, their correlation with malignancy and radioresistance, and the potential of mitochondrial transplantation to enhance radiation sensitivity to be of interest.

Your recognition of the significance of extracellular vesicles (EVs) in transferring genetic material, including mitochondria, and their influence on cancer cell behavior, particularly in the context of radioresistance and sensitivity, is encouraging.

Thank you for your suggestions. We have carefully reviewed and addressed all areas needing improvement before publishing (please see the attached file). We appreciate your feedback and will improve the manuscript to make it more valuable and relevant to the scientific community.

We look forward to incorporating your suggestions and resubmitting an improved version of the manuscript.

Reviewer 3 Report

Comments and Suggestions for Authors

In this study, Leonov et al. consider the role played by extracellular vesicles and mitochondria in radioresistance and radiosensitivity of Non-Small Cell Lung Cancer. The manuscript provides sufficient information on the role of mitochondria and extracellular vesicles in the development of lung cancer and beyond, as well as their methods of identification. The figures facilitate the reader's understanding the various cellular mechanisms considered. The manuscript is clear and well written and contains all the data of recent literature. I suggest a careful reading for typing errors. I wish the authors good luck in the publication process.

Minor revisions: 

Lane 53: MtO is reported just as abbreviation. Please, add the full name.

Lane 338: In Table 2 use capital letters at the beginning of each sentence.

Comments on the Quality of English Language

 Minor editing of English language required

Author Response

Dear Reviewer,

Thank you for your thoughtful and encouraging review of our manuscript. We appreciate your positive feedback regarding the clarity of our manuscript, as well as the comprehensive coverage of the role of mitochondria and extracellular vesicles in lung cancer radioresistance development and related mechanisms.

We are pleased that the figures included in the manuscript have aided in conveying the cellular mechanisms discussed effectively. Your suggestion regarding a careful proofreading for any typographical errors is duly noted, and we will ensure to address this in our final revisions.

We express our gratitude for your constructive feedback and good wishes for the publication process. Your input is invaluable to us, and we are committed to addressing any remaining concerns to ensure the quality and accuracy of our work.

Warm regards,

Minor revisions were resolved as follows: 

Lane 53: MtO is reported just as abbreviation. Please, add the full name.

Thank you very much for the note; we have moved forward the explanation of MtO from line 304 to line 23.

Lane 338: In Table 2 use capital letters at the beginning of each sentence.

Thank you very much for the note; we have capitalized the first letters in both Table 2 and Table 3. 

Reviewer 4 Report

Comments and Suggestions for Authors

In this manuscript, entitled " Extracellular Vesicles and Mitochondria Induced Radioresistance and Radiosensitivity of Non-Small Cell Lung Cancer" authors present at the beginning the use of radiotherapy for Non-Small Cell Lung Cancer (NSCLC) treatment and the existing problems (resistance and lack of biomarkers) of that therapy. Then, they analyzed further the resistance in cancer cells by presenting studies that investigated the mechanisms underlying the increased resistance. They highlight the study of EVs and its functions and how it can be useful for the understanding of cancer resistance and sensitivity. Moreover, authors present the implication of mitochondria dysfunction in cancer and how their transfer between the cells can affect the cancer progression.

I found the manuscript to be overall well written and be well described. The presented review will be useful to cancer field and biomarkers discovering science. However, I found few parts of the manuscript to be inadequate (see below).

Minor
Line 57 Please explain what means MtO/ MtS, I can not follow the flow of manuscript.

Fig.1 please write in figure legend what authors mean with the number 1, 2, 3

Fig.2 Please be more descriptive in fig.legend

Line 162-163 “It is well-known that the repair of DNA double-strand breaks (DSBs) induced by ionizing radiation in the CSCs population proceeds more efficiently.” Please add reference

Line164-166 “ In this regard, it is very important to investigate the spectrum of exosomal miRNAs secreted by irradiated cancer cells and their role in the regulation of the expression increased expression of proteins involved in EMT processes and DNA DSBS repair [24,25].” Please rephrase for more clarification

Fig.3 Authors have added numbers in order to explain the role of EV cargo, however it is not clear in which part each number is refer to. Authors should add numbers in the image in order to be easy to follow the text.

Fig.4 Similar response as Fig. 3

For the aforementioned reasons, I think that the manuscript should be published and can be further considered for publication in Cancers, only if the concerns be revised and clarified.

Comments on the Quality of English Language

It needs moderate correction of grammatical and spelling errors

Author Response

Thank you for taking the time to review our manuscript. We appreciate your positive feedback on the overall quality of the manuscript and its potential contribution to the cancer field and biomarker discovery science.

We acknowledge your feedback regarding certain inadequacies in the manuscript and assure you that we carefully addressed these concerns in our revisions, as described in the attached file. Your constructive criticism is invaluable to us, and we are grateful for your insights. We have made the necessary adjustments to improve the clarity and completeness of the manuscript, with the aim of enhancing its relevance and impact in the field. Our utmost dedication lies in guaranteeing the meticulous depiction and thorough presentation of every facet encompassing the study. Our aim is to deliver a holistic comprehension of the pivotal role played by extracellular vesicles and mitochondria in determining the radioresistance and radiosensitivity of NSCLC.

Once again, we appreciate your thoughtful review and look forward to incorporating your suggestions into our revised manuscript.

Round 2

Reviewer 1 Report

Comments and Suggestions for Authors

The review has been improved, but not sufficiently. The overall impression is that the content of the review is too far removed from the expectations of the title. Key points in the title are EVs, and mitochondria in the context of RS and RR in NSCLC. There seems to be little evidence for these topics in the literature, or they are not sufficiently emphasized by the authors. I wonder if the title is meant to be a suggestion by the authors based on some literature evidence on other cancers and limited evidence on lung cancer. If this is the case, the authors should consider thinking a different title. Almost all sections should focus on EVs and/or mitochondria involvement in non-small cell lung cancer onset/progression or in radiotherapy effects for non-small cell lung cancer. A fundamental problem is the lack of an introduction section in which the authors briefly describe the state-of-art of the argument of the title and clearly define the aim of the review and the modality they used in selecting the literature they relied on to write the article (e.g. the keywords). I consider this an important task and encourage authors to look at other review articles on similar arguments. Finally, I recommend summarising any information that is not relevant to the topics of the title to provide more clarity to the reader and to better substantiate the topic of the review.

Reviewer 2 Report

Comments and Suggestions for Authors

I appreciate that the authors have addressed all of my comments and concerns in the revised manuscript. The revised version improved significantly and deserved to be published in the esteemed journal Cancer in its present form.

Author Response

Thank you for taking the time to review the revised manuscript. We are pleased to hear that you appreciate the efforts made to address your comments and concerns. Your feedback has been invaluable in improving the quality of the manuscript. We are delighted that the revised version has met your expectations and that you consider it worthy of publication in Cancer. Your positive evaluation encourages us to believe that our work will make a meaningful contribution to the field. We are grateful for your support and constructive criticism throughout this process.

Round 3

Reviewer 1 Report

Comments and Suggestions for Authors

The authors stated: “We have taken great care to ensure that the manuscript thoroughly examines the roles of extracellular vesicles (EVs) and mitochondria in the context of radiosensitivity (RS) and radioresistance (RR) in non-small cell lung cancer (NSCLC). We understand that the literature available on these particular subjects may not be as extensive as in other fields. However, we have conducted thorough research and included pertinent studies that offer valuable insights into the role of EVs and mitochondria in NSCLC radioresponse. We have focused on the importance of these factors in affecting the treatment results of NSCLC, showing their potential as important targets for therapy.”

Authors must state in the introduction how many studies there are on NSCLC and EV/mitochondria and radioresistance/sensitivity (because if there is little evidence, the scientific literature on which they rely should be underlined). Authors need to emphasise this as a limitation and provide a scientific explanation as to why the information is so sparse.

About the title, the authors stated:” We respect your perspective on the title and its alignment with the content of the review. We understand your concern about the title and its potential implications. We chose the title based on our experience in the field of studying extracellular vesicles (EVs) and mitochondria and their role in radiosensitivity and radioresistance in non-small cell lung cancer (NSCLC). Although there may be limited literature on NSCLC compared to other cancers, it is essential to synthesize the available evidence and focus on ongoing research to advance our knowledge of NSCLC biology and therapy. As researchers with years of experience in studying the radioresistance of NSCLC and recently delving into the role of mitochondria in the radiosensitivity of radioresistant NSCLC cells, we feel an urgent need to disseminate our findings and insights through this review. Through this modest review, our primary goal is to uncover potential novel mechanisms that can be rigorously tested regarding NSCLC's response to radiation. By doing so, we will bridge the existing knowledge gap and stay at the forefront of cutting-edge research in this rapidly advancing field.”

I do not agree with their point of view. My criticism relates to the fact that the title must give the reader a rough idea of the content. If it is true that there is not much evidence for NSCLC in this area, the authors might agree that the title needs to be changed to something that shows that this is a challenge that needs to be addressed in more depth (e.g. “Extracellular Vesicles and Mitochondria Induced Radioresistance and Radiosensitivity of Non-Small Cell Lung Cancer: Future Challenges for Treatments?”).

If this is the case, the authors need to justify their aim in detail as a starting point for future investigations (if I have correctly understood what the authors want to achieve with this review).

The authors stated “We appreciate your suggestion regarding the introduction section. Thank you for your input! We have carefully considered your feedback and have incorporated an introduction section into the manuscript (lines 37-73). We are confident that the inclusion of this introduction effectively alleviates your concerns and offers essential context for readers. We hope that you will appreciate the enhanced comprehensiveness and informativeness of the revised manuscript. “

No references are given in the "Introduction" section. This must be supplemented accordingly. As mentioned above, the authors need to address the paucity of literature on NSCLC by providing references and the key challenges that this review aims to highlight for NSCLC.

Other criticisms:

The abstract contained no information about NSCLC and appears to be a summary of the introduction. The abstract needs to evidence the aim of the review and the importance of EV and mitochondria in radiotherapy for NSCLC.

 “Today cancer is now the third leading cause of global mortality [1].” Ref 1 is from 2008. I think it is a more recent publication, e.g. CA CANCER J CLIN 2021;71:209–249. All references need to be rechecked to see if they correspond to the latest information. 

Also, in the conclusions section, authors must add references to the works they refer to. In addition, in this section, authors must state their conclusions on EVs and future direction, and mitochondria and future direction. Finally, authors are encouraged to mention or comment on the interplay between EVs and mitochondria in radioresistance and radiosensitivity of NSCLC.

Finally, in the conclusion section, the authors should address whether there are clinical trials in this field that could also open up new perspectives for NSCLC.
